# Privacy concerns regarding personal health information in Myanmar: A cross-sectional survey in a least developed country

Khaing Zin Zin Htwe[1], Saranath Lawpoolsri[1,2], Ngamphol Soonthornworasiri[1], Panithee Thammawijaya[3], Jaranit Kaewkungwal[1,2]*

1 Department of Tropical Hygiene, Faculty of Tropical Medicine, Mahidol University, Bangkok, Thailand, 2 Faculty of Tropical Medicine, Center of Excellence for Biomedical and Public Health Informatics (BIOPHICS), Mahidol University, Bangkok, Thailand, 3 Department of Disease Control, Bureau of Epidemiology, Ministry of Public Health, Thailand

* jaranit.kae@mahidol.ac.th

## Abstract

Protecting personal health information has long been a key ethical concern in healthcare. With the rise of digital health systems, privacy concerns have intensified. In least developed countries like Myanmar, where digital literacy is low, public and governmental awareness of privacy issues is limited, and data protection laws are lacking, the health information privacy area remains largely unaddressed and understudied. This study, therefore, aimed to assess the level of privacy concerns related to personal health information in Myanmar and identify the factors associated with these concerns. A cross-sectional survey was conducted from March to May 2024 among 424 participants recruited online and through two private clinics. A validated 21-item scale measuring six domains: Collection, Errors, Unauthorized Secondary Use, Improper Access, Control, and Awareness, was used. Confirmatory factor analysis was performed to validate the scale, and logistic regression was used to identify factors associated with the concerns. Among the 424 participants, 77.83% reported high concerns. The highest concern was observed in the Awareness domain (mean 5.02, SD 1.65), while the Collection domain showed the lowest (mean 3.41, SD 1.44). Participants aged 45 years or older had significantly lower odds of high concerns compared to those aged 18–24 years, with an adjusted odds ratio (aOR) of 0.29 (95% CI 0.09-0.88). In contrast, those reporting good health (aOR 3.72, 95% CI 1.69-8.23) and high health concerns (aOR 5.26, 95% CI 3.02-9.45) had increased odds of high privacy concerns. These findings highlight the need for robust privacy safeguards, legal frameworks, and trust-building measures to support Myanmar's transition to digital health systems. Sampling via online channels and private clinics likely introduced an urban/educated bias, limiting generalizability. However, the findings may still provide insights relevant for other least developed countries facing

**Data availability statement:** All data analyzed during this study are available in supporting files.

**Funding:** The author(s) received no specific funding for this work.

**Competing interests:** The authors have declared that no competing interests exist.

**Abbreviations:** aOR, adjusted odds ratio; CFA, confirmatory factor analysis; CFI, comparative fit index; EMR, electronic medical records; LDC, least developed country; OR, odds ratio; PHI, personal health information; PHIPC, personal health information privacy concerns; SRMR, standardized root mean square residual; VIF, variance inflation factor.

similar challenges in balancing digital innovation with the protection of personal health information.

## Author summary

We examined the level of personal health information privacy concerns of people in Myanmar at a time when digital health tools are growing but legal protections and public awareness remain limited. From March to May 2024, we surveyed 424 adults online and in two private clinics. We asked about six kinds of concerns, related to the amount of personal health information collected, errors in the information, who can access the information, whether they are reused without permission, how much control people have over their own information, and how clearly rules are explained. More than three in four participants reported high overall concern. People were more worried about not knowing how their information would be used, and least worried about the act of collection itself. Older adults tended to be less concerned than younger adults, while those who felt in good health and those who were more worried about their health showed greater concern. Patterns also varied by sex, place of residence, type of occupation, and familiarity with electronic medical records. Our findings point to practical steps for Myanmar's digital health transition: communicate clearly about data practices, give people meaningful control, prevent misuse and improper access, and strengthen organizational and legal safeguards. These actions can build trust and support safe, more equitable health information systems in Myanmar and in similar settings.

## Introduction

### Background

Personal health information (PHI) encompasses any health-related data collected or shared by entities within or outside health information systems. In the past, healthcare services primarily involved healthcare professionals and patients only, placing the responsibility for PHI privacy solely on medical personnel. However, the inevitable change of health information systems into electronic systems and information exchange has expanded the volume of collected health data, giving rise to growing privacy issues [1]. As electronic medical records (EMRs) often contain sensitive and detailed personal information, and with the increasing risk of data breaches, safeguarding PHI privacy and security has become a critical concern [2]. A systematic review identified privacy and security as the second most frequently mentioned barrier to digital health implementation, as cited by both healthcare professionals and patients [3]. Moreover, privacy concerns are strong predictors of patient acceptance when new digital health technologies are introduced [4].

Privacy concerns are shaped by cultural values, trust in the institutions, and confidence in data handlers [5]. Socio-political beliefs and contextual norms influence individuals' risk-benefit perceptions regarding data exchange. Notably, trust in political institutions shapes views of surveillance technology: those with higher trust may see such systems as protective, while others may view them as threats to privacy [6]. In developing countries, privacy concerns are growing as PHI is increasingly shared among various stakeholders [7]. Among ASEAN (Association of Southeast Asian Nations) member states, only Cambodia and Myanmar have yet to implement laws specifically protecting medical and health data [8].

Regarding the current situation in Myanmar, the digital health index is lower than the regional average due to several constraints [9], such as inadequacy of electricity supply, poor internet connectivity, and heightened data sensitivity in a setting where security considerations overwhelm everything [10]. In the private sector, however, there have been success stories regarding EMR implementation [11] and telemedicine services during the COVID-19 pandemic [12]. Since the military coup in February 2021, healthcare operations have been disrupted, and surveillance practices have intensified. Reports have emerged of breaches involving individuals' PHI [13]. Although Myanmar's digital health system is still in its infancy, addressing public concerns about PHI privacy is essential. Proactively mitigating these concerns will be critical for the successful adoption of health information systems once national stability is restored.

## Domains of PHI privacy concerns

Over the past two decades, a widely used measurement scale for information privacy concerns has been developed and applied in both developed and developing countries. Originally designed for organizational practices and digital environments such as internet use, social networking, and mobile applications [14–18], the scale focused on four primary domains: Collection, Errors, Unauthorized Secondary Use, and Improper Access, modeled as correlated first-order factors [19,20]. This framework was later revised into a second-order construct to improve its conceptual structure [21]. Subsequently, the scale was adapted for healthcare settings, where terms such as "companies" were replaced with "healthcare providers," "healthcare entities," or "medical facilities" in studies on health information exchange and EMR adoption [22–24]. The scale was further expanded to include two additional domains: Control and Awareness, bringing the total to six domains [25,26].

These six domains comprehensively capture essential concerns about PHI privacy in healthcare environments [19,20,24]. Concerns about Collection are those related to the amount of PHI collected by healthcare facilities relative to the benefits provided. Concerns about Errors refer to those regarding potential mistakes in handling PHI, including accidental and intentional errors. Unauthorized secondary use involves the use of PHI for purposes other than those originally authorized by the individual. Improper access refers to unauthorized individuals gaining access to PHI. Control addresses whether individuals have control over their own PHI within healthcare facilities. Awareness measures individuals' knowledge of healthcare facilities' privacy practices regarding their PHI.

## Previous research on PHI privacy concerns

Studies evaluating PHI privacy concerns (PHIPC) using 7-point Likert scales across diverse settings have revealed varying outcomes influenced by country-specific contexts. For instance, an early study conducted in the United States reported relatively high concern levels in three of four key domains: Unauthorized secondary use (6.01), Improper access (5.95), Collection (5.22), and Errors (4.73) [27]. A study in New Zealand found moderate concern levels across the domains: Collection (3.79), Errors (4.13), Unauthorized secondary use (4.40), and Improper access (4.49) [28]. In South Africa, participants reported high concern levels in Collection (6.10), Control (6.00), and Awareness (6.40) [25]. In contrast, a study from Ghana showed high concerns in Errors (6.28), Unauthorized secondary use (6.01), and Improper access (6.37), but low concerns about Collection (3.36) [29]. Studies in Asia indicated moderate to low levels of concern. In Taiwan, the scores were Collection (3.54), Errors (4.09), Unauthorized secondary use (4.60), and Improper access (4.39) [24]. In Hong Kong, concerns across all six domains were moderate: Collection (4.27), Errors (4.33), Unauthorized secondary use (4.28), Improper access (4.61), Control (4.12), and Awareness (4.87) [26].

Several studies have investigated the antecedents of PHIPC, generally group them into five main categories: socio-demographic factors (such as demographic characteristics, personality traits, and personal knowledge and experience), social-relational factors, organizational and task environmental factors, macro-environmental factors, and information contingencies like the type and sensitivity of information [22–31]. In this study, we focused on eight socio-demographic predictors: age, sex, education, occupation, residence, EMR awareness, perceived health status, and health concerns. Prior research has demonstrated inconsistent findings regarding the influence of these factors on PHIPC [7,20,23,25,28,29,31].

Regarding age, studies in New Zealand, South Africa, Ireland, and the United States found that older individuals expressed greater privacy concerns. However, research from Ghana reported higher concerns among younger individuals, while a Taiwanese study observed no age-related differences. In terms of sex, findings also varied: studies in South Africa and Taiwan found no significant sex differences, whereas studies in the United States and Ghana indicated that males exhibited higher levels of concern. Education levels showed mixed results as well: higher privacy concerns were associated with lower education levels in South Africa, but the opposite was observed in Taiwan. In Ghana, no significant differences were found based on education. Regarding occupation, a Taiwan study found no occupational differences in privacy concerns, but individuals with EMR awareness demonstrated heightened concerns. Health status and health concerns also revealed diverse outcomes. Studies in Ireland reported no association between perceived health status and privacy concerns, whereas a U.S. study linked perceived poor health status to moderate concerns. In Ghana, individuals with higher health concerns showed greater privacy concerns.

These findings underscore the variability of PHIPC across different populations and settings, reflecting cultural, social, and systemic differences. In Myanmar, where public awareness of data privacy regulations is still emerging, concepts such as control, consent, and secondary use of health data may not be as widely recognized or emphasized as in other countries where the PHIPC framework has been tested. This highlights the importance of adapting and validating the PHIPC framework in the Myanmar context.

Myanmar is classified as a Least Developed Country (LDC), a status based on three primary criteria: per capita income, economic vulnerability, and human assets, which encompass indicators such as nutrition, health, education, and literacy. The United Nations postponed Myanmar's graduation from LDC status to 2027, citing setbacks from the 2021 military takeover [32]. To date, the country lacks a comprehensive data protection law, relying on limited provisions in existing legislation. For example, the 2008 Constitution, specifically Section 3 of the Law for Protection of Privacy, includes vague guarantees of personal privacy, while the Electronic Transactions Law (2004, amended in 2014 and 2021) focuses on electronic transactions and includes some privacy-related provisions. However, these protections are undermined by broad governmental authority that permits intervention under vague justifications such as "stability," "tranquility," and "national security" [33].

Myanmar's healthcare system faces profound challenges in service delivery and accessibility, which have worsened due to the combined impact of the COVID-19 pandemic and ongoing political instability since the February 2021 coup. These events have reduced service capacity and further eroded public trust, prompting a growing number of individuals to turn to alternative or private healthcare services. Additionally, the ongoing armed conflict has introduced severe logistical and operational barriers for healthcare delivery and health data collection, further complicating efforts to safeguard PHI and implement digital health initiatives [34].

## Objectives

The two primary objectives of this study were (1) to assess the level of PHIPC among Myanmar citizens and (2) to identify socio-demographic and health-related factors associated with high PHIPC.

## Methods

### Study design

We conducted a cross-sectional online survey to assess the level of PHIPC and associated factors among Myanmar citizens. A cross-sectional design was appropriate for estimating the prevalence of high PHIPC and identifying associated

socio-demographic and health-related factors at a single time point, particularly given the logistical and safety constraints posed by ongoing political instability.

## Setting

Data collection occurred between March and May 2024, during a period of ongoing political instability in Myanmar following the February 2021 military coup. However, no major political incidents that could affect data collection were reported during that period. The online survey link was distributed primarily via Facebook and Messenger, which are the most widely used social media platforms in Myanmar [35], and through two private outpatient clinics located in Kyaukpadaung and Pakokku townships. These clinics were selected to supplement online recruitment and reach individuals who may not be actively using social media, ensuring inclusion of individuals with lower digital literacy and limited social media engagement.

## Participants

Eligible participants were Myanmar citizens aged 18 years or older who provided informed consent to participate. A convenience sampling approach was used, combining online and clinic-based recruitment. Given the recruitment strategy and platform reliance, the sample may not be fully representative of Myanmar's general population.

## Study size

The sample size was calculated using the formula for estimating infinite population proportions [36], assuming 50% of Myanmar citizens had high PHIPC, with a 95% confidence level:

$$n = (z^2 \times p \times q)/d^2$$

where:

$$z = 1.96,$$
$$p = 0.5,$$
$$q = 1 - p = 0.5,$$
$$d = 0.05.$$

This yielded a required minimum sample size of 385 participants. A total of 424 complete responses were ultimately collected and analyzed. The finite population correction was not applied.

## Assessments

The primary outcome was PHIPC, measured using a validated scale covering six domains: Collection, Errors, Unauthorized Secondary Use, Improper Access, Control, and Awareness. Each domain contained 2–3 items rated on a 7-point Likert scale (1 = strongly disagree to 7 = strongly agree), with higher scores indicating greater privacy concerns. At the beginning of the questionnaire, a brief explanation of PHI, based on the Health Insurance Portability and Accountability Act (HIPAA), was provided to ensure respondents understood the concept of PHI.

Independent variables included age, sex, residence (urban or rural), education, occupation, EMR awareness, perceived health status, and health concerns. Perceived health status was self-assessed by participants using a 7-point Likert scale ranging from "poor" to "well," reflecting their overall subjective evaluation of health. Health concerns were similarly self-rated on a 7-point Likert scale from "not concerned" to "very concerned," indicating the extent to which individuals were concerned about their current health status.

## Translation process

The translation and cultural adaptation of the questionnaire followed established international guidelines [37–39]. A multi-step forward-backward translation process was conducted to ensure linguistic accuracy, conceptual equivalence, and cultural relevance of the PHIPC scale.

In the forward translation phase, two independent bilingual translators, fluent in English and Burmese and familiar with Myanmar's healthcare context, translated the original English version into Burmese. Their translations were compared and synthesized into a single preliminary version by a reviewer. For the back translation phase, two additional bilingual translators with healthcare experience translated the synthesized Burmese version back into English. These two translators had never seen the original English version before or during the translation process, nor did they know who the forward translators were. Another reviewer compared and reconciled those backward-translated versions into only one. This back-translated version was compared to the original to identify any semantic discrepancies, cultural mismatches, or conceptual inconsistencies. Any discrepancies were resolved collaboratively by the previous two reviewers, who both are bilingual and have experience with Myanmar's healthcare setting. This preliminary Burmese version was pilot-tested with a convenience sample of 32 participants recruited via Messenger. Participants were asked to provide feedback on comprehension and ease of understanding. Based on their input, minor wording adjustments were made to improve while retaining the intended meaning of each item.

## Instrument validation

The PHIPC scale was adapted from a previously validated instrument [26], which has been tested to be relevant in the health information setting by previous studies [40,41]. To validate the conceptual model of PHI privacy in the developed questionnaire, confirmatory factor analysis (CFA) with a second-order construct was conducted to evaluate factor loadings and model fit indices. Goodness-of-fit was assessed based on Hair et al.'s guideline, suggesting acceptable fit when Comparative Fit Index (CFI) > 0.92 with Standardized Root Mean Square Residual (SRMR) ≤ 0.08 [42]. The reliability of the questionnaire was assessed using Cronbach α to determine internal consistency for each domain and the overall PHIPC.

## Data analysis

Responses from Google Forms were exported as an XLSX file and analyzed using RStudio (version 2024.12.0 + 467). Demographic information was summarized descriptively. Perceived health status and health concerns, measured in a 7-point Likert scale, were classified into two groups: high level (>4) and low level (≤4). PHIPC scores were calculated as average scores of all domains and categorized into two groups: high concerns (>4) and low concerns (≤4). This dichotomization approach was selected to facilitate the practical interpretation of predictors relevant to health policy decisions and to account for the bounded distribution of Likert-based scores. Preliminary analyses using continuous PHIPC scores yielded similar results, supporting the robustness of the dichotomous approach. Additionally, comparative model diagnostics further justified dichotomization, with the logistic regression model demonstrating substantially better model fit (AIC = 391.26; BIC = 452.01) compared to linear regression using continuous PHIPC scores (AIC = 1378.02; BIC = 1442.81). The association between antecedent factors and the PHIPC (overall and domain-specific) was analyzed using logistic regression. Crude odds ratios (ORs), adjusted odds ratios (aORs), and corresponding 95% confidence intervals were calculated. To further explore whether these associations varied across key subgroups, we conducted stratified analyses by residence (urban vs. rural) and education level (bachelor's degree or higher vs. below bachelor). These subgroup analyses were informed by notable sample imbalances: our study population had a disproportionately higher proportion of urban and highly educated participants compared to the general population of Myanmar. By performing stratified logistic regressions, we aimed to assess whether patterns of privacy concerns differed within more representative subpopulations. There was no missing data because all fields were set as required to be filled in the Google Form.

**Ethical considerations**

Ethical approval was granted by the Ethics Committee of the Faculty of Tropical Medicine, Mahidol University (MUTM 2024-007-01). Participants were informed of detailed study information in the introduction of the online survey and were required to provide electronic consent before proceeding to the questionnaire. To ensure anonymity, no personally identifiable information, including names, email addresses, or IP addresses, was collected. The data platform was secured and accessible only to authorized investigators. These procedures also helped safeguard participants during ongoing political instability in Myanmar by ensuring that no sensitive or identifying information was collected. No compensation was provided to participants.

## Results

### Characteristics of study participants

As shown in Table 1, out of 424 usable responses, 59.2% (251/424) of the respondents were female, and 75.47% (320/424) resided in urban areas. The largest age group was 25–31 years (44.58%, 189/424). Regarding education, 61.08% (259/424) held a bachelor's degree or higher. In terms of occupation, 56.37% (239/424) were employed in non-healthcare sectors, 27.59% (117/424) in healthcare-related roles, and 16.04% (68/424) were unemployed. On a 7-point scale measuring perceived health status, 90.33% (383/424) rated their health as "well" (scores >4). Similarly, for health concerns (1 = not concerned, 7 = very concerned), 57.31% (243/424) were classified as having high health concerns (scores >4). Regarding EMR awareness, 44.34% (188/424) reported understanding EMRs, 33.96% (144/424) had heard of EMRs but did not fully understand them, and 21.7% (92/424) had never heard of EMRs.

### Conceptual model validation of PHIPC

The six-domain structure of PHIPC was validated using CFA with a second-order factor model. As shown in Fig 1, the first-order model demonstrated good fit across all six domains: Collection, Errors, Unauthorized Secondary Use, Improper Access, Control, and Awareness. Most items yielded standardized factor loadings above the recommended threshold of 0.70, except for Col1 (0.56) and Err3 (0.68), which showed slightly lower loadings within acceptable limits [42].

In the second-order model, each domain demonstrated strong loadings onto the higher-order PHIPC construct, ranging from 0.71 to 0.98, except for Collection (0.44). The overall model fit indices indicated good model fit, with CFI of 0.93 and SRMR of 0.06, meeting recommended cutoffs for acceptable fit.

The reliability of the overall PHIPC scale was high, as indicated by a Cronbach α of 0.93. The reliability indices for the individual domains were also satisfactory: Collection (0.75), Errors (0.80), Unauthorized secondary use (0.88), Improper access (0.86), Control (0.79), and Awareness (0.80).

### Levels of PHIPC

As presented in Table 2, the overall mean (SD) score of PHIPC across all domains was 4.64 (1.25). Most domains had average scores ranging between 4.5 and 5.0, indicating moderate concern levels. The Collection domain was an exception, with a lower mean score of 3.41 (1.44). A cut-off score of ≤4 was used to define "low concerns," and >4 to indicate "high concerns." Based on this classification, 77.83% (330/424) of participants exhibited high overall PHIPC.

When analyzed by domain, over 70% of participants expressed high concerns regarding Unauthorized Secondary Use, Improper Access, Control, and Awareness. Concern levels were slightly lower for the Errors domain, with 64.62% (274/424) reporting high concerns. Notably, only 24.53% (104/424) reported high concerns in the Collection domain. Fig 2 displays the distribution of PHIPC scores for each domain using histograms. Table 3 provides the mean scores for each item within the six PHIPC domains.

PLOS Digital Health

**Table 1. Profile of study participants (n = 424).**

| | Frequency | Percentage |
|---|---|---|
| **Age Group** | | |
| 18-24 years | 74 | 17.45 |
| 25-31 years | 189 | 44.58 |
| 32-38 years | 89 | 20.99 |
| 39-45 years | 35 | 8.25 |
| >45 years | 37 | 8.73 |
| **Sex** | | |
| Female | 251 | 59.2 |
| Male | 169 | 39.86 |
| Prefer not to say | 4 | 0.94 |
| **Residence** | | |
| Rural | 104 | 24.53 |
| Urban | 320 | 75.47 |
| **Education** | | |
| Bachelor's degree or higher | 259 | 61.08 |
| Below bachelor's degree | 165 | 38.92 |
| **Occupation** | | |
| Healthcare | 117 | 27.59 |
| Non-healthcare | 239 | 56.37 |
| Unemployed | 68 | 16.04 |
| **EMR Awareness** | | |
| Have not heard | 92 | 21.7 |
| Have heard, but not understand | 144 | 33.96 |
| Understand | 188 | 44.34 |
| **Perceived Health Status (from poor to well in 7-point Likert scale)** | | |
| Poor (1–4) | 41 | 9.67 |
| Well (5–7) | 383 | 90.33 |
| **Health Concerns (from not concerned to concerned in 7-point Likert scale)** | | |
| Not concerned (1–4) | 181 | 42.69 |
| Concerned (5–7) | 243 | 57.31 |

## Factors associated with PHIPC

Table 4 presents proportions of high PHIPC and Table 5 presents the associations between antecedent factors and high PHIPC scores overall, as well as stratified by residence and education level. Variance inflation factor (VIF) testing indicated no multicollinearity between predictor variables (VIF < 3). Overall, compared to individuals aged 18–24 years, those aged 45 years and older had lower adjusted odds of reporting high PHIPC (aOR 0.29, 95% CI 0.09-0.88). Perceived health status and health concerns showed robust associations with PHIPC. Individuals perceiving their health status as "well" had significantly higher adjusted odds of reporting high PHIPC (aOR 3.72, 95% CI 1.69-8.23). Likewise, those with higher health concerns had greater odds of high PHIPC (aOR 5.26, 95% CI 3.02-9.45). These associations are visually summarized in Fig 3. The forest plot presents both crude and adjusted ORs with corresponding 95% CIs on a logarithmic scale.

Subgroup analyses indicated variation in factors associated with high PHIPC scores by residence and education levels (Table 5). Among urban residents, lower odds of high PHIPC were observed in individuals aged 25–31 years (aOR 0.24,

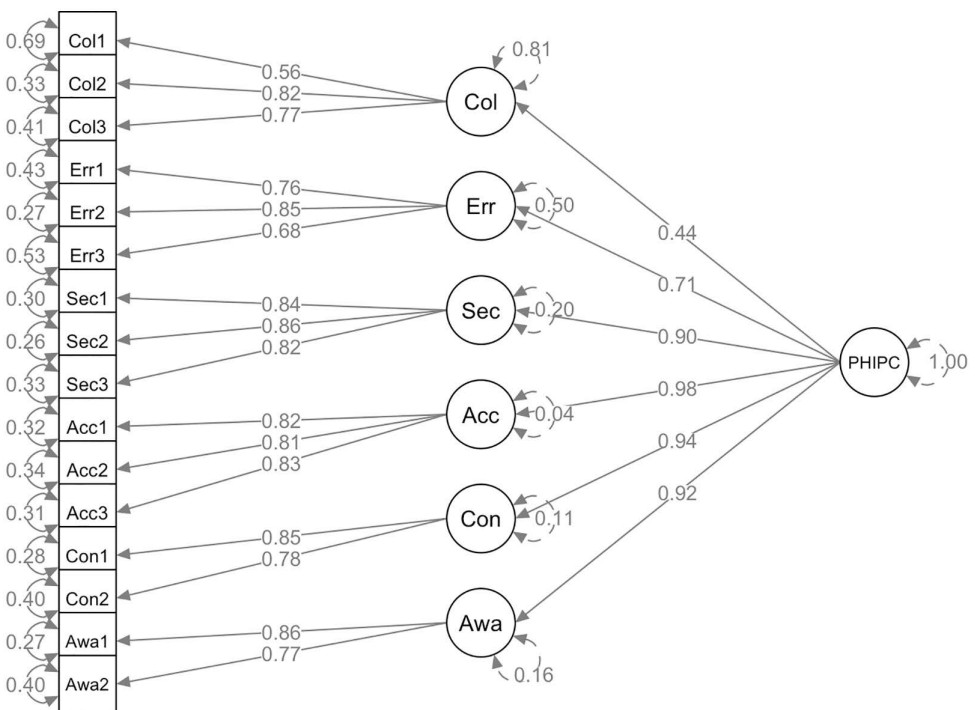

**Fig 1. PHIPC model.** Note: Col: Collection; Err: Errors; Sec: Unauthorized secondary use; Acc: Improper access; Con: Control; Awa: Awareness.

**Table 2. Levels of PHIPC (n = 424).**

| PHIPC Domains | Mean (SD) | Low Concerns | High Concerns |
|---|---|---|---|
| | | n (%) | n (%) |
| Overall | 4.64 (1.25) | 94 (22.17) | 330 (77.83) |
| Collection | 3.41 (1.44) | 320 (75.47) | 104 (24.53) |
| Errors | 4.55 (1.44) | 150 (35.38) | 274 (64.62) |
| Unauthorized secondary use | 4.90 (1.68) | 105 (24.76) | 319 (75.24) |
| Improper access | 4.98 (1.60) | 95 (22.41) | 329 (77.59) |
| Control | 5.01 (1.67) | 106 (25) | 318 (75) |
| Awareness | 5.02 (1.65) | 101 (23.82) | 323 (76.18) |

95% CI 0.06-0.82) and 32–38 years (aOR 0.16, 95% CI 0.04-0.60) compared to those aged 18–24 years. In rural settings, having an education below a bachelor's degree significantly increased odds of high PHIPC (aOR 27.00, 95% CI 2.51-659.00), although this estimate was imprecise as reflected by the wide CI. Males with below bachelor's degree education also demonstrated significantly higher odds of reporting high PHIPC (aOR 3.88, 95% CI 1.13-15.90). In the group of participants with lower than bachelor's degree, those who had heard of EMRs but did not understand them had higher odds of PHIPC compared to those who had never heard of EMRs (aOR 4.53, 95% CI 1.03-19.70). Consistent findings persisted regarding health status (for all subgroups except urban and bachelor's degree or higher) and health concerns (for all subgroups), reinforcing their importance as robust predictors of PHIPC.

Additional analyses by individual PHIPC domains (Tables S2-S7 in S1 Table) revealed domain-specific variations in associations. For the Collection domain, participants who reported health concerns had significantly lower odds of

PLOS Digital Health

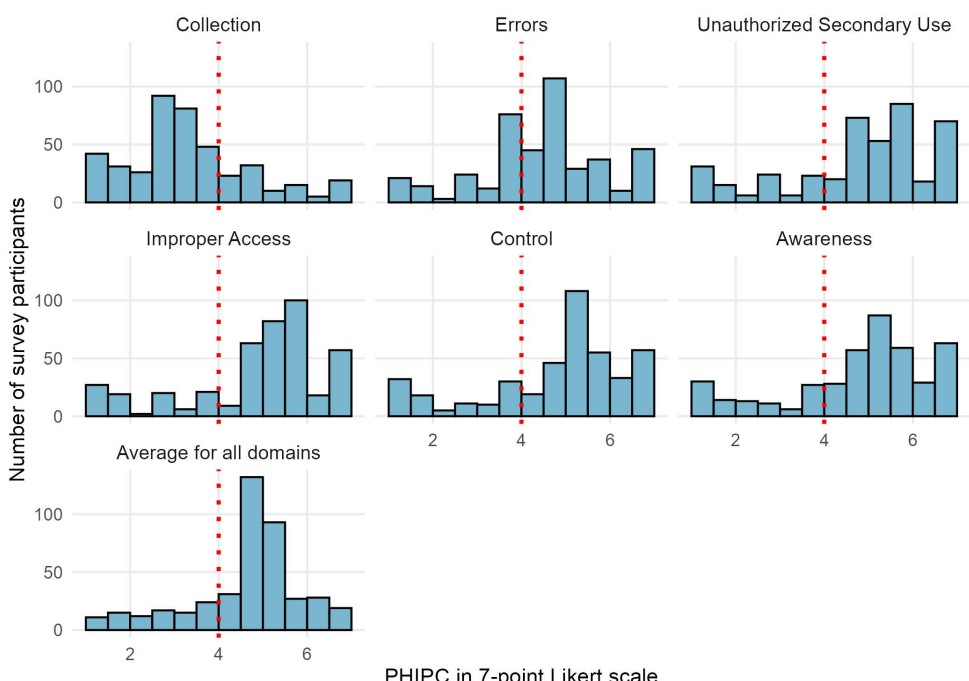

**Fig 2. Distribution of PHIPC.**

expressing high privacy concerns (aOR 0.46, 95% CI 0.28-0.74). Regarding the Errors domain, urban residents exhibited significantly higher odds of concern (aOR 2.30, 95% CI 1.28-4.17), whereas those with partial EMR awareness (having heard but not understood) showed lower odds of concern (aOR 0.40, 95% CI 0.20-0.76). In the Unauthorized Secondary Use domain, significant associations were identified for males (aOR 1.93, 95% CI 1.12-3.39), participants older than 45 years (aOR 0.26, 95% CI 0.09-0.74), those with good perceived health status (aOR 2.94, 95% CI 1.36-6.38), and individuals with high health concerns (aOR 4.71, 95% CI 2.80-8.13). For the Improper Access domain, significant predictors included being older than 45 years (aOR 0.35, 95% CI 0.12-0.98), good perceived health status (aOR 2.44, 95% CI 1.11-5.29), and high health concerns (aOR 4.80, 95% CI 2.80-8.46). Within the Control domain, participants reporting good health status (aOR 2.86, 95% CI 1.29-6.36) and high health concerns (aOR 6.15, 95% CI 3.61-10.80) had significantly greater odds of expressing concerns. Lastly, the Awareness domain showed significant associations with employment status and health-related variables: unemployed individuals (aOR 4.41, 95% CI 1.54-13.70), non-healthcare workers (aOR 2.03, 95% CI 1.03-4.09), those with good health status (aOR 4.04, 95% CI 1.82-9.06), and those reporting high health concerns (aOR 6.87, 95% CI 3.93-12.50) demonstrated significantly higher odds of expressing privacy concerns.

## Discussion

We gathered data from individuals recruited online and from two private clinics in Myanmar, obtaining 424 valid responses. The participant demographic was relatively specific: approximately two-thirds (320/424) resided in urban areas, around 60% (259/424) held a bachelor's or higher degree, and about one-fourth of the participants (117/424) worked in the healthcare sector. Despite the limited implementation of EMR systems in Myanmar, only about 20% of participants (92/424) reported being unfamiliar with EMRs, possible because the survey was distributed to healthcare personnel and patients visiting outpatient clinics.

**Table 3. Mean scores of each item in the six domains of PHIPC.**

|  | Item description | Mean (SD) |
|---|---|---|
| **Collection** | | |
| Col1 | It usually bothers me when healthcare entities ask me for personal health information. | 3.34 (1.83) |
| Col2 | It bothers me to give personal health information to so many healthcare entities. | 3.37 (1.72) |
| Col3 | I'm concerned that healthcare entities are collecting too much personal health information about me. | 3.53 (1.72) |
| **Errors** | | |
| Err1 | I am concerned that healthcare entities do not take enough steps to make sure that my personal health information in their files is accurate. | 4.28 (1.71) |
| Err2 | I am concerned that healthcare entities do not have adequate procedures to correct errors in my personal health information. | 4.57 (1.72) |
| Err3 | I am concerned that healthcare entities do not devote enough time and effort to verifying the accuracy of personal health information in their databases. | 4.80 (1.66) |
| **Unauthorized secondary use** | | |
| Sec1 | I'm concerned that when I give personal health information to a healthcare entity for some reason, the entity would use the information for other reasons. | 4.80 (1.88) |
| Sec2 | I am concerned that healthcare entities would sell my personal health information in their databases to other entities. | 4.86 (1.93) |
| Sec3 | I am concerned that healthcare entities would share my personal health information with other entities without my authorization. | 5.03 (1.83) |
| **Improper access** | | |
| Acc1 | I am concerned that healthcare entities do not devote enough time and effort in preventing unauthorized access to my personal health information. | 4.97 (1.83) |
| Acc2 | I am concerned that healthcare entities' databases that contain my personal health information are not protected from unauthorized access. | 4.94 (1.80) |
| Acc3 | I am concerned that healthcare entities do not take enough steps to make sure that unauthorized people cannot access my personal health information in their databases. | 5.02 (1.78) |
| **Control** | | |
| Con1 | It usually bothers me when I do not have control or autonomy over decisions about how my personal health information is collected, used and shared by healthcare entities. | 5.01 (1.80) |
| Con2 | It usually bothers me when I do not have control of personal health information that I provide to healthcare entities. | 5.01 (1.87) |
| **Awareness** | | |
| Awa1 | It usually bothers me when healthcare entities seeking my personal health information do not disclose the way the data were collected, processed, and used. | 5.03 (1.80) |
| Awa2 | It usually bothers me when I am not aware or knowledgeable about how my personal health information will be used by healthcare entities. | 5.01 (1.82) |

The level of PHIPC in Myanmar was found to be moderate, with average scores ranging from 4 to 5 on a 7-point Likert scale across most domains, except Collection, which scored below 4. This suggests that while individuals in Myanmar were less concerned about the collection of PHI, they placed greater emphasis on its management, particularly regarding

**Table 4. Proportions of participants with high PHIPC scores (>4).**

| | All participants[a] (n=330) | Residence | | Education | |
| --- | --- | --- | --- | --- | --- |
| | | Rural[a] (n=84) | Urban[a] (n=246) | Bachelor's degree or higher[a] (n=199) | Below bachelor's degree[a] (n=131) |
| **Age Group** | | | | | |
| 18-24 years | 65 (87.84) | 23 (85.19) | 42 (89.36) | 7 (87.5) | 58 (87.88) |
| 25-31 years | 150 (79.37) | 27 (81.82) | 123 (78.85) | 125 (79.62) | 25 (78.13) |
| 32-38 years | 65 (73.03) | 17 (77.27) | 48 (71.64) | 41 (68.33) | 24 (82.76) |
| 39-45 years | 30 (85.71) | 10 (90.91) | 20 (83.33) | 17 (100) | 13 (72.22) |
| >45 years | 20 (54.05) | 7 (63.64) | 13 (50) | 9 (52.94) | 11 (55) |
| **Sex** | | | | | |
| Female | 187 (74.50) | 54 (79.41) | 133 (72.68) | 109 (72.67) | 78 (77.23) |
| Male | 142 (84.02) | 30 (85.71) | 112 (83.58) | 89 (83.18) | 53 (85.48) |
| Prefer not to say | 1 (25) | 0 (0) | 1 (33.33) | 1 (50) | 0 (0) |
| **Residence** | | | | | |
| Rural | 84 (80.77) | | | 12 (63.16) | 72 (84.71) |
| Urban | 246 (76.88) | | | 187 (77.92) | 59 (73.75) |
| **Education** | | | | | |
| Bachelor's degree or higher | 199 (76.83) | 12 (63.16) | 187 (77.92) | | |
| Below bachelor's degree | 131 (79.39) | 72 (84.71) | 59 (73.75) | | |
| **Occupation** | | | | | |
| Healthcare | 83 (70.94) | 8 (80) | 75 (70.09) | 72 (71.29) | 11 (68.75) |
| Non-healthcare | 190 (79.5) | 47 (77.05) | 143 (80.34) | 121 (81.76) | 69 (75.82) |
| Unemployed | 57 (83.82) | 29 (87.88) | 28 (80) | 6 (60) | 51 (87.93) |
| **EMR Awareness** | | | | | |
| Have not heard | 77 (83.7) | 49 (87.50) | 28 (77.78) | 9 (64.29) | 68 (87.18) |
| Have heard, but not understand | 105 (72.92) | 24 (70.59) | 81 (73.64) | 54 (70.13) | 51 (76.12) |
| Understand | 148 (78.72) | 11 (78.57) | 137 (78.74) | 136 (80.95) | 12 (60) |
| **Perceived Health Status** | | | | | |
| Poor | 20 (48.78) | 3 (27.27) | 17 (56.67) | 13 (68.42) | 7 (31.82) |
| Well | 310 (80.94) | 81 (87.1) | 229 (78.97) | 186 (77.5) | 124 (86.71) |
| **Health Concerns** | | | | | |
| Not concerned | 110 (60.77) | 14 (46.67) | 96 (63.58) | 76 (63.33) | 34 (55.74) |
| Concerned | 220 (90.53) | 70 (94.59) | 150 (88.76) | 123 (88.49) | 97 (93.27) |

[a]presented as "n (%)".

errors, misuse, and access controls. These findings align with studies conducted in New Zealand, Taiwan, and Hong Kong, but contrast with higher levels of concern reported in the United States, South Africa, and Ghana [24,26–29]. Some studies indicated that national contexts, such as socio-political systems, cultural beliefs, and the availability of data privacy laws, shape PHIPC [5–7], resulting in variations across countries. For example, a study during the COVID-19 pandemic found a much higher general willingness to share data in China compared to the United States and Germany [5]. Yet, our study reveals that the levels of PHIPC among Myanmar's population, even in its current LDC status and under instability, were not significantly different from those in countries with contrasting national contexts.

**Table 5. Adjusted odds ratio (aORs) and 95% CIs for high PHIPC scores among all participants and subgroups stratified by residence and education.**

| | All participants[a] | Residence | | Education | |
| --- | --- | --- | --- | --- | --- |
| | | Rural[a] | Urban[a] | Bachelor's degree or higher[a] | Below bachelor's degree[a] |
| **Age Group** | | | | | |
| 18-24 years | Reference | Reference | Reference | Reference | Reference |
| 25-31 years | 0.56 (0.20, 1.48) | 4.69 (0.38, 77.60) | 0.24 (0.06, 0.82)* | 0.59 (0.03, 4.02) | 0.43 (0.08, 2.07) |
| 32-38 years | 0.39 (0.13, 1.08) | 4.61 (0.29, 107.00) | 0.16 (0.04, 0.60)** | 0.26 (0.01, 1.91) | 0.70 (0.12, 4.16) |
| 39-45 years | 1.23 (0.33, 5.05) | 2.01 (0.10, 76.40) | 0.57 (0.11, 3.04) | 7635591.00 (0, Inf) | 0.39 (0.06, 2.56) |
| >45 years | 0.29 (0.09, 0.88)* | 3.63 (0.18, 125.00) | 0.13 (0.03, 0.48)** | 0.30 (0.01, 2.64) | 0.23 (0.04, 1.22) |
| **Sex** | | | | | |
| Female | Reference | Reference | Reference | Reference | Reference |
| Male | 1.57 (0.89, 2.80) | 3.40 (0.38, 42.30) | 1.61 (0.86, 3.07) | 1.49 (0.73, 3.11) | 3.88 (1.13, 15.9)* |
| Prefer not to say | 0.33 (0.01, 3.01) | 0.00 | 0.48 (0.02, 5.94) | 0.88 (0.03, 23.9) | 0.00 |
| **Residence** | | | | | |
| Rural | Reference | | | Reference | Reference |
| Urban | 0.97 (0.45, 2.03) | | | 2.17 (0.64, 7.06) | 0.62 (0.20, 1.86) |
| **Education** | | | | | |
| Bachelor's degree or higher | Reference | Reference | Reference | | |
| Below bachelor's degree | 0.96 (0.46, 2.06) | 27.00 (2.51, 659.00)* | 0.55 (0.24, 1.26) | | |
| **Occupation** | | | | | |
| Healthcare | Reference | Reference | Reference | Reference | Reference |
| Non-healthcare | 1.70 (0.87, 3.39) | 0.07 (0.00, 1.52) | 1.92 (0.94, 4.00) | 2.18 (1.00, 4.97) | 0.41 (0.05, 2.66) |
| Unemployed | 1.73 (0.63, 4.99) | 0.09 (0.00, 7.44) | 1.46 (0.44, 5.23) | 0.68 (0.15, 3.36) | 0.92 (0.10, 8.24) |
| **EMR Awareness** | | | | | |
| Have not heard | Reference | Reference | Reference | Reference | Reference |
| Have heard, but not understand | 0.67 (0.30, 1.49) | 2.05 (0.19, 31.7) | 0.64 (0.22, 1.79) | 1.37 (0.31, 5.89) | 0.99 (0.28, 3.48) |
| Understand | 1.34 (0.52, 3.43) | 1.27 (0.08, 26.90) | 1.33 (0.42, 3.97) | 4.53 (1.03, 19.70)* | 0.31 (0.06, 1.68) |
| **Perceived Health Status** | | | | | |
| Poor | Reference | Reference | Reference | Reference | Reference |
| Well | 3.72 (1.69, 8.23)** | 103.00 (9.23, 2417.00)*** | 2.36 (0.89, 6.08) | 0.94 (0.25, 3.21) | 31.3 (7.60, 179.00)*** |
| **Health Concerns** | | | | | |
| Not concerned | Reference | Reference | Reference | Reference | Reference |
| Concerned | 5.26 (3.02, 9.45)*** | 55.10 (7.13, 1008.00)** | 3.95 (2.12, 7.62)*** | 4.44 (2.20, 9.38)*** | 14.7 (4.34, 69.20)*** |

* $P$ < .05, ** $P$ < .01, *** $P$ < .001.

a presented as "aOR (95% CI)".

Recent publicized data breaches and surveillance developments in Myanmar likely heightened public awareness of certain privacy risks. By 2024, the military's expanded surveillance infrastructure under the pretense of e-government development included building a national e-ID system with extensive and detailed biometric data [43]. This political context likely contributed to increased concerns in domains such as Unauthorized Secondary Use and Improper Access, even if Collection concerns remained lower.

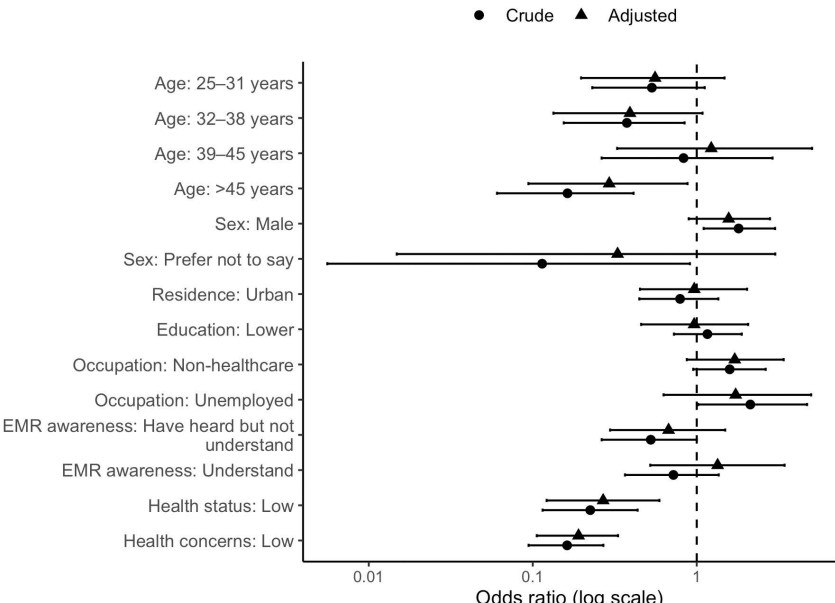

**Fig 3. Forest plot of crude and adjusted ORs with 95% CIs for factors associated with high PHIPC.**

Many studies highlight that high PHIPC can significantly hinder the development and adoption of digital health systems [44–48]. Conversely, individuals with lower levels of privacy concerns may consent to use technologies without fully understanding their privacy rights. This could result in a lower demand for comprehensive privacy protection measures, leaving them vulnerable to potential misuse of their data [49,50]. Research has generally found a strong link between PHIPC and trust in health information technologies, emphasizing the need to integrate core privacy principles and employ trusted design features [47,48,51]. However, there is no universally optimal level of PHIPC to ensure the successful implementation of digital health systems. It remains uncertain whether the moderate PHIPC levels observed in Myanmar will act as a barrier or a facilitator for developing future digital health systems in the country.

In this study, perceived health status and health concerns were significantly associated with PHIPC in several domains: individuals with greater health concerns or those who perceived themselves as healthier tended to exhibit higher levels of PHIPC. These findings are consistent with some previous studies [29,52,53]. However, other studies found no significant differences in privacy concerns based on general health condition or perceived health status [48,54,55]. Sociodemographic variables exhibited domain-specific association: older individuals showed lower concerns in Unauthorized Secondary Use and Improper Access, males expressed higher Unauthorized Secondary Use concerns, urban residents had higher Errors concerns, and non-healthcare and unemployed individuals had higher Awareness concerns. EMR awareness influenced the Errors domain uniquely, with partially aware individuals reporting lower concerns. This finding contradicts a previous study that suggested familiarity with EMRs increased privacy concerns [23]. This could be attributed to the fact that some participants in this study, being healthcare providers, were somewhat familiar with the technology, which may have fostered trust in the system and its data quality. A previous survey conducted in Myanmar found that healthcare staff in non-governmental clinics viewed EMRs as useful and intended to adopt them [11]. Additionally, another study highlighted that patients are more likely to share information when they trust the accuracy and confidentiality of their electronic records. In contrast, privacy concerns can lead to withholding information or delaying treatment [56]. This emphasizes the importance of building trust in EMR systems through strong controls and ensuring data integrity [20,52,57].

Subgroup analyses highlighted notable demographic variations. Younger urban individuals had lower privacy concerns compared to their rural counterparts, and rural residents with education below a bachelor's degree exhibited particularly elevated concerns. These results emphasize the vulnerability of certain demographic groups linked to limited digital literacy and highlight the importance of targeted interventions to address privacy concerns. Previous literature also shows varying results depending on the setting and different healthcare technologies. A systematic review of 33 studies on patients' perspectives regarding the confidentiality, privacy, and security of collected data revealed both conflicting and aligned views, influenced by factors like age, income, marital status, and experience with health technologies [58]. Some studies found no significant differences in PHIPC based on sex, age, and education level [25,31,54,55,59] while others found associations with age and education [7,23,29,57].

## Implications

These findings carry several implications for the development of health information systems in Myanmar, both in the public and private sectors. Although a National Health Information System had been planned for 2017–2021 [60], its progress has effectively halted due to the political instability following the 2021 military coup. As such, our findings are most immediately applicable to the design and governance of health data systems in the private sector, where digital health initiatives continue to expand despite national-level stagnation. However, these insights may significantly inform future efforts to revitalize the National Health Information System once political conditions stabilize. In particular, the relatively low level of concern reported in the Collection domain highlights the need to raise public awareness of privacy risks related to data capture, whether led by government or private entities.

## Limitations

This was the first study conducted in Myanmar to address information privacy in the healthcare sector. As a partially online survey conducted during a period of national instability, the study has several limitations. First, the participant sample was not fully representative of Myanmar's population as a result of convenience sampling. Specifically, our recruitment approach likely over-represented urban, more educated, and digitally connected participants. Therefore, the prevalence of high PHIPC and the observed associations may not generalize to rural or lower-literacy populations. Accordingly, we present these results as exploratory and hypothesis-generating, reflecting privacy concerns among healthcare-accessing populations rather than the Myanmar population as a whole. While over 70% of Myanmar's citizens reside in rural areas and only 7.3% have completed a bachelor's degree [61], this study included 25% rural residents and 61% participants with bachelor's degrees. However, in line with literature recommendations to establish baseline insights for the design, development, and implementation of systems in developing countries [62–64], the study targeted internet users while also collecting data from healthcare providers, patients, and their relatives in clinical settings. Despite its underrepresentation, the study encompassed a broader sociodemographic spectrum of stakeholders critical to future system development. Although the sample size was not specifically calculated for subgroup analysis, we conducted it to explore potential differences, given that the distribution of study participants differed notably from that of the target population. While we acknowledge the limitations in statistical power, the exploratory subgroup analysis adds valuable context and may help generate hypotheses for future research.

Second, the concept of privacy in Myanmar does not readily translate into the local language or cultural framework [65]. With low digital literacy levels, limited privacy awareness in public and governmental sectors, and a lack of effective data protection laws [65,66], ensuring respondents understood privacy protections posed challenges. To address this, key terms were explained at the start of the questionnaire. Third, the survey gathered a limited range of potential influencing factors. A significant finding, perceived health status, was self-reported and may not accurately reflect reality. Future research exploring the relationship between actual health status and perceptions of PHIPC is recommended. Moreover, due to the voluntary nature of participation and the online survey format, response bias cannot be entirely ruled out.

Individuals who are more interested in privacy topics or who are more digitally literate may have been more likely to participate. Additionally, since recruitment channel was not captured, we cannot quantify participation by social media versus clinics, limiting detailed assessment of channel-specific selection bias.

Lastly, the moderate PHIPC level observed may fluctuate given the study's timing during political instability. Myanmar has faced ongoing cybersecurity issues, including regular hacking of government websites and cases of identity theft and privacy breaches during the COVID-19 pandemic [65,66]. Thus, the PHIPC levels reported may only represent the conditions at the time and could change under different circumstances. Additionally, the post-2021 coup context may have shaped who could safely participate in an online survey. Intermittent internet disruptions and heightened safety concerns could have reduced participation among individuals with unstable connectivity or those avoiding online engagement for security reasons, contributing to additional selection bias. This context may also have influenced responses by increasing caution or sensitivity regarding data sharing and trust, although these factors were not directly measured in this study. As such, unmeasured confounding cannot be entirely ruled out. We also recommend that future studies employ representative sampling and to study the association of PHIPC with the success/failure of implementation of digital health information systems in LDCs, including Myanmar.

## Conclusions

This study explored PHIPC in Myanmar, an LDC with limited public awareness of data privacy and the absence of comprehensive data protection laws. Drawing from a diverse sample, our findings revealed moderate levels of privacy concerns, particularly in domains related to data management, namely Errors, Unauthorized Secondary Use, Improper Access, Control, and Awareness, while concerns regarding data collection were comparatively lower. The most influential factors associated with high PHIPC were perceived health status and health concerns. Individuals who rated their health as good or expressed greater concern for their health were significantly more likely to report high levels of privacy concern. Although overall PHIPC was not significantly influenced by demographic factors such as age, sex, education level, residence, or EMR awareness, subgroup and domain-specific analyses indicated that these variables may play a role in shaping concerns within specific domains of PHIPC. While the moderate concern levels observed may not present a direct barrier to the rollout of digital health initiatives in Myanmar, the findings underscore the importance of building public trust and establishing robust privacy safeguards. To ensure the successful adoption and long-term sustainability of digital health systems, policymakers and developers must prioritize clear privacy policies, transparent communication, and secure data management practices. These considerations are not only critical for Myanmar but also offer valuable insights for other nations navigating similar digital health transformations under constrained political or infrastructural conditions.

## Supporting information

**S1 Data. De-identified participant-level dataset.** This file contains two sheets. In the sheet "Form Responses" (n = 424; 29 variables; one row per participant), there are demographics and covariates, screening items (citizenship and consent in Burmese language), and PHIPC item responses on a 1–7 scale. In the sheet "Coding", we provide category coding and item/domain mapping used in analyses.
(XLSX)

**S1 Table. Additional regression tables.**
(DOCX)

**S1 File. Questionnaire.** Survey questionnaires in English.
(PDF)

**S2 File. Questionnaire.** Survey questionnaires in Myanmar.
(PDF)

## Acknowledgments

The authors extend their gratitude to the Faculty of Tropical Medicine, Mahidol University, for supporting this research project. Portions of the manuscript preparation involved the use of generative AI (ChatGPT by OpenAI, Grammarly, and Elicit) to assist with language editing, phrasing refinement, literature search, and providing coding assistance during R script development for data analysis. All AI-generated content was carefully reviewed, edited, and validated by the authors to ensure scientific accuracy, integrity, and compliance with ethical standards. No AI tools were used for the conceptualization, translation, data collection, or generation of study results.

## Author contributions

**Conceptualization:** Khaing Zin Zin Htwe, Saranath Lawpoolsri, Ngamphol Soonthornworasiri, Jaranit Kaewkungwal.

**Data curation:** Khaing Zin Zin Htwe, Jaranit Kaewkungwal.

**Formal analysis:** Khaing Zin Zin Htwe, Jaranit Kaewkungwal.

**Investigation:** Khaing Zin Zin Htwe.

**Methodology:** Khaing Zin Zin Htwe, Saranath Lawpoolsri, Ngamphol Soonthornworasiri, Jaranit Kaewkungwal.

**Software:** Khaing Zin Zin Htwe.

**Supervision:** Saranath Lawpoolsri, Ngamphol Soonthornworasiri, Panithee Thammawijaya, Jaranit Kaewkungwal.

**Validation:** Khaing Zin Zin Htwe, Jaranit Kaewkungwal.

**Visualization:** Khaing Zin Zin Htwe, Jaranit Kaewkungwal.

**Writing – original draft:** Khaing Zin Zin Htwe, Saranath Lawpoolsri, Ngamphol Soonthornworasiri, Panithee Thammawijaya, Jaranit Kaewkungwal.

**Writing – review & editing:** Khaing Zin Zin Htwe, Jaranit Kaewkungwal.

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
