## [Decision Letter · Decision Letter 0]

5 Feb 2026

Response to Reviewers '. This file does not need to include responses to any formatting updates and technical items listed in the 'Journal Requirements' section below.* A marked-up copy of your manuscript that highlights changes made to the original version. You should upload this as a separate file labeled 'Revised Manuscript with Track Changes '.* An unmarked version of your revised paper without tracked changes. You should upload this as a separate file labeled 'Manuscript '. If you would like to make changes to your financial disclosure, competing interests statement, or data availability statement, please make these updates within the submission form at the time of resubmission. Guidelines for resubmitting your figure files are available below the reviewer comments at the end of this letter. We look forward to receiving your revised manuscript. Kind regards, J. Mark Ansermino, MBBChSection EditorPLOS Digital Health Leo Anthony CeliEditor-in-ChiefPLOS Digital Healthorcid.org/0000-0001-6712-6626  **Journal Requirements:**

1. Please upload separate figure files in .tif or .eps format. Also, remove the figures from your manuscript file but keep the legends.

2. We note that your Data Availability Statement is currently as follows: All data analyzed during this study are available in supporting files.

**Additional Editor Comments (if provided):** The reviewers have made a conssitent assesment of your important manuscript. I look forward to reviewing the revised version. I apologize for the slow reposnse but it would seem that the original Academic Editor has been unresponsive.**Reviewers' Comments:** Reviewer's Responses to Questions

**Comments to the Author**

1. Does this manuscript meet PLOS Digital Health’s publication criteria ? Is the manuscript technically sound, and do the data support the conclusions? The manuscript must describe methodologically and ethically rigorous research with conclusions that are appropriately drawn based on the data presented.

Reviewer #1: Yes

Reviewer #2: Yes

Reviewer #3: Yes

2. Has the statistical analysis been performed appropriately and rigorously?

Reviewer #1: Yes

Reviewer #2: Yes

Reviewer #3: Yes

3. Have the authors made all data underlying the findings in their manuscript fully available (please refer to the Data Availability Statement at the start of the manuscript PDF file)?

Reviewer #1: Yes

Reviewer #2: Yes

Reviewer #3: Yes

4. Is the manuscript presented in an intelligible fashion and written in standard English?

Reviewer #1: Yes

Reviewer #2: Yes

Reviewer #3: Yes

Reviewer #1: This study provides meaningful and original insight into digital health privacy concerns. Its findings have practical implications for policymakers, developers, and researchers interested in digital trust and ethical health data use.

To further strengthen the manuscript, I suggest the authors:

Emphasize the sampling limitations (urban and educated bias) more prominently in the Abstract and Limitations section.

Clarify the translation validation process (like how conceptual equivalence was verified beyond back-translation).

Provide exact details on the data files shared in the supplementary material.

Consider a few language refinements for readability.

Reviewer #2: Overall Assessment: The manuscript addresses a timely and understudied topic. Privacy concerns around personal health information are critical in least developed countries like Myanmar, especially during the shift to digital health systems. The study fills an important gap. It uses a validated scale and provides clear findings on high concern levels and associated factors. The results are relevant for policy in Myanmar and similar settings. However, the sampling method limits representativeness. Reporting of methods and results needs more detail. The work is solid but requires revisions for clarity and rigor.

Strengths: The topic is highly relevant. Few studies exist on health data privacy in least developed countries. The use of a validated 21-item scale with six domains is appropriate. Confirmatory factor analysis adds credibility. The high response rate (77.83% with high concerns) and clear domain differences are interesting. Associations with age, self-reported health, and health worries are meaningful. Implications for trust-building and legal frameworks are practical.

Major Concerns:

1) The sample uses convenience methods. Recruitment happened mainly via Facebook and two private clinics. Participants are mostly urban (75%), female (59%), and well-educated (61% bachelor’s degree or higher). Many work in healthcare (28%). Younger adults are over-represented. This group likely has higher digital literacy and privacy awareness. Rural, older, less-educated, and low-income people are under-represented. Findings cannot speak for all of Myanmar. The authors must stress this limitation and present results as exploratory for accessible populations.

2) Political instability after the 2021 coup is mentioned briefly. Internet restrictions and safety fears probably affected who could join online. No major incidents occurred during data collection, but the context still matters. Expand discussion on how the situation may influence responses and access.

3) Regression results show wide confidence intervals in some subgroups. This reflects small subgroup sizes. Interpretation should stay cautious.

4) Details on recruitment are missing. How many were invited online versus in clinics? These points affect reproducibility and bias assessment.

5) The manuscript uses only tables for results. This works but feels dense. Other visualizations would make patterns easier to grasp.

Recommendations:

Can you compare the statistics among social media vs clinic statistics? The scenario could be different and can be good addition to the paper. Acknowledge sampling limitations explicitly. Frame the study as exploratory among accessible urban and clinic-based adults. Add recruitment details (numbers approached, response rates by method). Discuss the political and digital access context in limitations. Strengthen the discussion of implications for rural and marginalized groups.

Reviewer #3: This manuscript addresses an important and timely topic in digital health by examining personal health information privacy concerns in a low-resource (LDC). Overall, the manuscript is clear and informative, with well-presented results supported by organized tables and figures. The discussion effectively places the findings within existing literature, and the conclusions are appropriately drawn from the results. The authors also demonstrate transparency by clearly reporting study limitations and providing practical recommendations for stakeholders. The effort to align with open science principles by indicating how underlying data and tables can be accessed is appreciated.

The comments below are mainly minor and focus on improving clarity, consistency, and organization. Greater alignment between the abstract and the Objectives section would strengthen the manuscript. The main objective stated in the abstract should be clearly reflected as the primary objective throughout the paper, with consistent use of terminology (e.g., using either “extent” or “level” of privacy concerns consistently). If both primary and secondary objectives are intended, these should be clearly separated and simply stated.

In the Methods section, minor reorganization would improve readability. The Study Design subsection could briefly explain why a cross-sectional online survey is appropriate for achieving the study aim, rather than restating the objective. Separating the setting, data collection, recruitment, and sampling strategy into distinct subsections would also enhance clarity. In addition, details related to instrument validation (such as confirmatory factor analysis) would be more appropriately placed in the Data Collection Tool/Instruments section rather than in the Data Analysis section.

A few additional clarifications would further improve accessibility. Naming the private clinics involved in participant recruitment, instead of referring only to cities, would improve transparency. Defining abbreviations at first mention or moving the abbreviations list earlier in the manuscript would also enhance readability. Finally, the manuscript would benefit from a brief section explicitly highlighting the study’s key contributions to the field.

Overall, these are minor, addressable points. Addressing them would further strengthen the clarity and organization of an already solid and valuable contribution to digital health research.

what does this mean?If published, this will include your full peer review and any attached files.). If published, this will include your full peer review and any attached files.

**Do you want your identity to be public for this peer review?** For information about this choice, including consent withdrawal, please see our Privacy Policy .

Reviewer #1: **Yes:** Hana AbbasianHana Abbasian

Reviewer #2: No

Reviewer #3: **Yes:** Josiane AKINGENEYEJosiane AKINGENEYE

**Figure resubmission:** While revising your submission, we strongly recommend that you use PLOS’s NAAS tool (https://ngplosjournals.pagemajik.ai/artanalysis) to test your figure files. NAAS can convert your figure files to the TIFF file type and meet basic requirements (such as print size, resolution), or provide you with a report on issues that do not meet our requirements and that NAAS cannot fix.

**Reproducibility:** To enhance the reproducibility of your results, we recommend that authors of applicable studies deposit laboratory protocols in protocols.io, where a protocol can be assigned its own identifier (DOI) such that it can be cited independently in the future. Additionally, PLOS ONE offers an option to publish peer-reviewed clinical study protocols. Read more information on sharing protocols at https://plos.org/protocols?utm_medium=editorial-email&utm_source=authorletters&utm_campaign=protocols To enhance the reproducibility of your results, we recommend that authors of applicable studies deposit laboratory protocols in protocols.io, where a protocol can be assigned its own identifier (DOI) such that it can be cited independently in the future. Additionally, PLOS ONE offers an option to publish peer-reviewed clinical study protocols. Read more information on sharing protocols at https://plos.org/protocols?utm_medium=editorial-email&utm_source=authorletters&utm_campaign=protocols

---

## [Editor Report · Decision Letter 1]

9 Mar 2026

Privacy concerns regarding personal health information in Myanmar: A cross-sectional survey in a least developed country

PDIG-D-25-00646R1

Dear Dr. Kaewkungwal,

We are pleased to inform you that your manuscript 'Privacy concerns regarding personal health information in Myanmar: A cross-sectional survey in a least developed country' has been provisionally accepted for publication in PLOS Digital Health.

Best regards,

Laura Sbaffi, PhD, MA, MSc

Section Editor

PLOS Digital Health